# Delivery of CRISPR/Cas9 Plasmid DNA by Hyperbranched Polymeric Nanoparticles Enables Efficient Gene Editing

**DOI:** 10.3390/cells12010156

**Published:** 2022-12-30

**Authors:** Kemao Xiu, Laura Saunders, Luan Wen, Jinxue Ruan, Ruonan Dong, Jun Song, Dongshan Yang, Jifeng Zhang, Jie Xu, Y. Eugene Chen, Peter X. Ma

**Affiliations:** 1Department of Biologic and Materials Sciences, University of Michigan, Ann Arbor, MI 48109, USA; 2Macromolecular Science and Engineering Center, University of Michigan, Ann Arbor, MI 48109, USA; 3Center for Advanced Models and Translational Sciences and Therapeutics, University of Michigan, Ann Arbor, MI 48109, USA; 4Department of Biomedical Engineering, University of Michigan, Ann Arbor, MI 48109, USA; 5Department of Materials Science and Engineering, University of Michigan, Ann Arbor, MI 48109, USA

**Keywords:** nanoparticle, gene delivery, pDNA, CRISPR/Cas9, PEI, polyplex, gene editing

## Abstract

Gene editing nucleases such as CRISPR/Cas9 have enabled efficient and precise gene editing in vitro and hold promise of eventually achieving in vivo gene editing based therapy. However, a major challenge for their use is the lack of a safe and effective virus-free system to deliver gene editing nuclease elements. Polymers are a promising class of delivery vehicle due to their higher safety compared to currently used viral vectors, but polymers suffer from lower transfection efficiency. Polymeric vectors have been used for small nucleotide delivery but have yet to be used successfully with plasmid DNA (pDNA), which is often several hundred times larger than small nucleotides, presenting an engineering challenge. To address this, we extended our previously reported hyperbranched polymer (HP) delivery system for pDNA delivery by synthesizing several variants of HPs: HP-800, HP-1.8K, HP-10K, HP-25K. We demonstrate that all HPs have low toxicity in various cultured cells, with HP-25K being the most efficient at packaging and delivering pDNA. Importantly, HP-25K mediated delivery of CRISPR/Cas9 pDNA resulted in higher gene-editing rates than all other HPs and Lipofectamine at several clinically significant loci in different cell types. Consistently, HP-25K also led to more robust base editing when delivering the CRISPR base editor “BE4-max” pDNA to cells compared with Lipofectamine. The present work demonstrates that HP nanoparticles represent a promising class of vehicle for the non-viral delivery of pDNA towards the clinical application of gene-editing therapy.

## 1. Introduction

The landscape of gene editing has changed in recent years with the advent of Zinc Finger Nuclease (ZFN), Transcription Activator-Like Effector Nuclease (TALEN), and CRISPR (clustered regularly interspaced short palindromic repeats)/Cas9 (CRISPR Associated Protein 9) technologies [1]. Nuclease based gene editing systems hold promise for clinical gene therapy. Several clinical trials have been launched utilizing ZFN (e.g., ClinicalTrials.gov Identifier: NCT00842634), TALEN (e.g., NCT03057912) and Cas9 (e.g., NCT03545815).

Cas9 is a promising gene editing nucleases given its ease of use, multiplex capacity, and unparalleled targeting efficiency [2,3,4]. Many groups including us have reported efficient Cas9 mediated gene editing in cultured cells [5,6] and animal embryos [7,8,9] for translational biomedical applications [10,11]. However, at least three major challenges remain as roadblocks for implementation of clinical gene editing therapeutics. First, off target effects, especially those associated with CRISPR/Cas9 need to be minimized or eliminated [12]. Second, the homology directed repair (HDR) rate remains low even with the help of gene editing nucleases, limiting the precision of correction based therapeutics [9]. The third and perhaps most difficult challenge is to realize an efficient and safe in vivo delivery method of gene editing elements, a hurdle for all nucleic acid therapeutics [13,14].

Issues inherent to current virus-based delivery method include the immune response to the vector itself, oncogenicity due to insertional mutagenesis, the expensive and limited scalability associated with manufacturing, and the relatively small DNA packaging capacity [15,16]. There have been resultant cases of cancer and death in ongoing clinical trials [17,18], highlighting the safety concerns associated with viral vectors, and prompting an increased interest in the development and utilization of non-viral delivery systems. However, non-viral delivery systems (such as liposomes and polymers) still suffer from low efficacy, although it is generally believed that they have better safety profiles than their viral counterparts [19,20,21]. Although liposome and polymer based gene delivery has made significant progress in in vitro studies and has received deserved attention during the past two decades, these non-viral gene delivery systems are still facing challenges when applied in vivo such as rapid clearance from blood, toxicity, and low efficiency [22,23]. A vast improvement in safety and efficiency for in vivo gene delivery systems is required for clinical application [24,25,26]. 

Polymer nanoparticles such as cationic particles, self-assembling particles, and polymer nanospheres [27] are an attractive non-viral means for nucleic acid delivery due to their ease of chemical modification, reproducibility, scalability, large capacity for packaging, and increased biosafety [13,24,28]. Previously, we developed a hyperbranched polymer (HP) system for microRNA (miRNA) delivery by attaching polyethylene glycol (PEG) and cationic polyethylenimine (PEI) chains to the outer shell of a hyperbranched polyester molecular core [29]. This HP system was demonstrated to be safe and efficient in packaging and delivering miRNA both in vitro and in vivo. Given the engineering flexibility of the HP, we hypothesized that it could be adapted and optimized for enhanced nucleic acid loading capacity to enable packaging and delivering pDNAs encoding gene editing nucleases which are several hundred times larger than miRNAs. Herein, we developed HP polymers bearing different molecular weight polyethyleneimine (PEI) chains (800, 1800, 10K, or 25K Da) and investigated their gene packaging abilities, cytotoxicity, and transfection efficiency. We report that HP-25K nanoparticles can be used to safely and efficiently package and deliver various pDNAs, including those encoding CRISPR/Cas9 and its variant, BE4-max, for gene editing applications. 

## 2. Materials and Methods

### 2.1. Polymer Synthesis

Polyethylenimine (PEI, Mw 800 Da and 25 kDa), polyethylene glycol methyl ether (PEG, Mw 2000 Da), and hyperbranched bis-MPA polyester (64 hydroxyl groups) were purchased from Sigma-Aldrich. PEI (Mw 1800 Da) was purchased from Alfa Aesar (Ward Hill, MA). PEI (Mw 10 kDa) was purchased from Polysciences Inc. (Warrington, PA, USA). All HP polymers (HP4-800, HP4-1.8K, HP4-10K, HP4-25K) were synthesized using methods previously described [29]. 

### 2.2. HP Solution Preparation 

HP was dissolved in DNase and RNase free distilled water (ultrapure water, Invitrogen) in 1–10 mg/mL concentration, and sonicated for 30 min, then centrifuged at 12,000 *g* for 5 min. The supernatant was used for all HP related experiments.

### 2.3. Plasmid and Cas9 Construct Preparation

CRISPR plasmids pX330 (#42230) and pX458 (#48138) and EGFP reporter plasmid (#40768) were obtained from Addgene. The mCherry expression plasmid pmCherry-N1 (#632523) was obtained from Clontech. The pDNA was extracted with a “QIAGEN Plasmid Maxi Kit” and the A260/280 ratio was about 1.8.

### 2.4. Particle Size and Zeta Potential Measurements

The particle size and zeta potential were measured using a Beckman Coulter DelsaNano C Submicron Particle Size Analyzer at room temperature (25 °C). The nanoparticles at various N/P ratios (nitrogen atoms of the polymer to phosphates of DNA) were prepared by slowly adding 60 pmol of DNA (pmcherry-N1 plasmids or pX458 plasmid DNA) solution to an appropriate volume of polymer solution (1.0 mg/mL) and were then incubated at room temperature for 30 min. The HP-pDNA nanoparticles were then diluted with Milli-Q water to a volume of 2.0 mL before measurement.

### 2.5. Gel Retardation Assay

The HP packaging efficiency was analyzed using a gel retardation assay. Polyplex solution (10 μL) was mixed with 2 μL of 6× loading buffer and loaded on an Ethidium bromide containing 1% agarose gel with Tris-acetate (TAE) running buffer (pH 8.0) and was electrophoresed at 120 V for 20 min. DNA bands were visualized with an ultraviolet (254 nm) illuminator and photographed with a BioSpectrum Imaging System (USA).

### 2.6. TEM Study on Morphology and Structure of HP-pDNA Nanoparticles

Polyplex solutions were imaged using a negative staining method previously developed in our lab to visualize the pDNA in this work. Tungsten-incorporated pDNA was prepared for use in the nanoparticles. Briefly, 10 mL of sodium tungstate aqueous solution (0.15 mol/L) was added dropwise to 6 mL of HCl aqueous solution (0.8 mol/L) at room temperature. The precipitate was collected and washed to obtain the activated tungstic acid. The desired quantity of activated tungstic acid was added to an aqueous DNA solution with a W/P molar ratio of 3. The W-incorporated pDNA solution was purified by centrifugation and stored at −80 °C. One drop of aqueous polyplex solution (1.0 mg/mL) was added to a carbon-coated copper grid. The grid was dried under ambient conditions. A JEOL 1400Plus TEM was used in this study to obtain TEM images.

### 2.7. Guide RNAs

Guide RNA (gRNA) target sequences were designed using an online tool at Custom Alt-R CRISPR-Cas9 guide RNA tool provided by Integrated DNA Technologies (IDT, https://www.idtdna.com/ (accessed on 29 November 2022)). 

The gRNA sequences are: 

TERT sg1: 5′-ACAATCGGCCGCAGCCCGTCAGG-3′; 

TERT sg2: 5′-CGCGTACGACACCATCCCCCAGG-3′; 

MYLIP sg: 5′-TCTGTACAATGCTGGCGTTGTGG-3′. 

PDCD1 sg: 5′-TTAGGGCAGGGCAGGCCGAGGGG-3′.

The PCSK9 gRNA was described in a previous report [30].

PCSK9 sg: 5′-GGTGCTAGCCTTGCGTTCCGAGG-3′.

The gRNA sequences were cloned in pX330 and pX458 plasmids for transfection experiments as previously described [31,32]. 

### 2.8. Cell Viability Assay

Cells: Ad293 cells (Catalog# 240085) were obtained from Agilent. HepG2 cells (Catalog# HB-8065) and Hela cells (Catalog# CCL-2) were obtained from ATCC. Ad293, HepG2 and Hela cells were cultured in Dulbecco’s Modified Eagle Medium (DMEM, Catalog#11965092, ThermoFisher, Waltham, MA, USA) supplemented with 10% fetal bovine serum (FBS, Catalog# SH30071.03HI, Hyclone, Logan, UT, USA) and 1% penicillin-streptomycin (Catalog# 15140122, ThermoFisher). AML-12 cells (Catalog# CRL-2254) were purchased from ATCC, and cultured in DMEM:F12 (Catalog# 11320033, ThermoFisher) supplemented with 10%FBS, 1% insulin-transferrin-selenium (ITS, Catalog# 41400045, ThermoFisher), 40 ng/mL dexamethasone (Catalog# D4902, Sigma-Aldrich, St. Louis, MO, USA), and 1% penicillin-streptomycin(Catalog# 15140122, ThermoFisher). Rabbit fibroblast cells were established in house from ear biopsies collected from New Zealand White Rabbits. The ear biopsy procedure is approved by the University of Michigan Institutional Animal Care and Use Committee (IACUC) protocol #PRO00010094. Rabbit fibroblast cells were cultured in DMEM (Catalog#11965092, ThermoFisher) supplemented with 10% FBS (Catalog# SH30071.03HI, Hyclone), and 1% penicillin-streptomycin (Catalog# 15140122, ThermoFisher).

The cytotoxicity of the polymer was assessed using an MTT assay kit (#CT02, Millipore, Burlington, MA, USA). 1 × 10^4^ rabbit embryonic fibroblast (REF) cells or HepG2 cells were seeded in a 96-well plate one day before encapsulation in the HPs. The amount of HP added to each well was calculated according to 200 ng plasmid DNA multiplied by the N/P ratio. HPs were added to each well accordingly and there were three replicates for each group. The MTT assay was carried out following the manufacturer’s manual.

### 2.9. Cellular Uptake Assay

Cells (Ad293 (a derivative of HEK293), human (HepG2) or mouse (AML12)) were seeded in a 24-well plate one day before the assay. Upon transfection, 1 µg of pDNA and polymer (HPs or PEIs) at the desired N/P ratio (*w*/*w*) were diluted in 50 µL of Opti-MEM (#31985062, Invitrogen, Waltham, MA, USA), incubated at room temperature for 30 min, and subsequently added into each well of cell culture. The control (lipofectamine) transfection process was following the manufacture’s protocol. Briefly, 50 µL lipofectamine dilution in Opti-MEM and 50 µL DNA dilution in Opti-MEM were mixed and incubated for 15 min. Then, the mixture was added into the well of cell culture. The transfection result was obtained after 24 h. 

### 2.10. Flow Cytometry

Cells were washed with PBS and treated with 0.25% trypsin. Cells were then transferred to a 15 mL conical centrifuge tube and spun down at 200 *g* for 5 min. The pellet cells were re-suspended in 2% FBS in PBS and filtered with a 70 μm Nylon cell strainer (#08-771-2, Falcon, Sydney, Australia). Flow cytometry was conducted at the University of Michigan Flow Cytometry Core.

### 2.11. Confocal Imaging

Cells were seeded in Chambered Cell Culture Slides (#08-774-25, Falcon), and plasma staining was carried out following the manufacturer’s instructions for CellMask Deep Red Plasma membrane stain (C10046, Thermo Fisher Scientific, Waltham, MA, USA). After removal from the chamber, the slide was mounted in prolong Gold anti-fade mount medium with DAPI (P36931, Thermo Fisher Scientific). Confocal images were taken in the University of Michigan microscopy and image analysis Laboratory with a Nikon-A1 confocal microscope.

### 2.12. Cas9 Induced Mutation Analysis and Quantification

Cas9 induced indel mutations at the target site were analyzed with a T7 endonuclease I (T7E1) assay or by deep sequencing (Deepseq) as we previously described [6]. The targeted mutation site was amplified via PCR and the PCR products were purified and used for the T7E1 assay. An amplicon of less than 280 bp was amplified before it was sent for deepseq sequencing.

Primers used to amplify target regions for T7E1 are listed below:

TERT sg1 

Forward 5′-GCTTCCCCCTAGTCTGTTGTCTGG-3′; 

Reverse 5′-CTGGCCCGGCTGCTTCTTGTGGTC-3′.

TERT Sg2 

Forward 5′-CCTGACTGCCCGGGCTCCTATT-3′; 

Reverse 5′-ACCTCCACCACAGAAACGCATCAC-3′.

PCSK9: 

Forward: 5′-CACGGCCTCTAGGTCTCCT-3′; 

Reverse: 5′-GCCTCCCATCCCTACACC-3′.

 

Primers used to amplify target regions for Deepseq are listed below:

TERT sg1 

Forward 5′-GGGGCTCAAACGCACTTCT-3′; 

Reverse 5′-ACGTCCAGACTCCGCTTCAT-3′.

TERT Sg2 

Forward 5′-CGTGAACCTTACGTGGCTCTT-3′; 

Reverse 5′-ACTCACACAGGTGGATGTGAC-3′.

MYLIP 

Forward 5′-AGGAGAAGCTACGCAAGCTG-3′; 

Reverse 5′-AGGAGGGATAGGTGAGGCTG-3′.

PCSK9: 

Forward: 5′-CACGGCCTCTAGGTCTCCT-3′; 

Reverse: 5′-GCCTCCCATCCCTACACC-3′.

PCSK9 primers amplified amplicons were used for both the T7 endonuclease assay and CRISPR-seq.

### 2.13. Cas9 Base Editor BE4-Max and gRNA Targeting PDCD1

Cas9 base editor BE4-max plasmid was obtained from Addgene (#112093). 

Primers for amplification of targeted region: 

Forward 5′-CTTCATCAGGGACTTAGCCTGGC-3′; 

Reverse 5′- CAGCCTGGTGCTGCTAGTCTG-3′.

### 2.14. Statistics

Data are expressed as mean ± standard error of means (SEM) from three replicates in bar graphs, and were analyzed and compared using unpaired, 2-tailed Student’s *t* test (Graphpad Prism 9.2.0, San Diego, CA, USA). Statistical significance with *p* < 0.05 is considered significant. Different levels of significant statistical differences were indicated by number of * in each graph, where * *p* < 0.05, ** *p* < 0.01, and *** *p* < 0.001.

## 3. Results and Discussion

### 3.1. HPs Efficiently Package Plasmid DNA In Vitro

PEI is a central component of the HP system and is one of the most studied cationic polymers for nucleic acid delivery due to its favorable DNA-condensing capabilities and ability to deliver pDNA in vitro and in vivo [25,26,27]. We previously developed an HP (HP-800) consisting of small molecular weight PEI (800 Da) for effective miRNA delivery [15]; however, HP-800 was not able to result in pDNA expression when it was used to transfect cells (data not shown). In the present work, we increased the PEI MW from 800 Da (HP-800) to 1.8 kDa (HP-1.8K), 10 kDa (HP-10K) and 25 kDa (HP-25K), while maintaining the same hyperbranched core and PEG composition (Figure 1A). Increasing the PEI MW increases the loading capacity of the vehicle and charge of the HP, creating a larger, more cationic nanoparticle for the pDNA to electrostatically combine with. Subsequent self-assembly into the HP-pDNA polyplex is driven by these electrostatic interactions [15]. The hydrophilic PEG chains are expected to form a stealth layer to protect the inner polyplexes from bioactive fouling and degradation (Figure 1B). The polyplexes formed a double-shell architecture of approximately 20 nm diameter with the pDNA tightly embedded inside, as shown by transmission electron microscopy (TEM) (Appendix A).

The zeta potentials of HP-800, HP-1.8K and HP-25K are lower than the PEIs of the same MW (PEI-800, PEI-1.8K, and PEI-25K) (Figure 1C), leading us to expect that cytocompatibility of the HPs will be higher than for PEI of a corresponding length. The diameters of HP nanoparticles HP-800, HP-1.8K, and HP-25K became smaller after they formed polyplexes with pDNA (Figure 1D). The observation that the size of the complex is smaller than that of the HP polymer is in good agreement with the literature [33]. Interestingly, HP-10K did not behave like other HP nanoparticles; HP-10K nanoparticles are of similar zeta potentials as PEI-10K, and sizes of HP-10K-pDNA polyplexes are like HP-10K alone. In summary, all HP/pDNA complex showed the particle size of less than 150 nm.

We next investigated whether HPs with different PEI chain lengths would be able to package pDNA in vitro, as assessed with a gel retardation assay using a reporter pDNA (pmCherry-N1 4.7 kb or pX458 9.3 kb). HPs and pDNAs were mixed at different N/P ratios and used in the electrophoresis-based gel retardation assay to determine whether the HPs could package various pDNA. The original HP-800 was found unable to package pmCherry-N1 (4.7 kb) effectively at N/P ratios at 1, as indicated by a clear band running away from the loading well (Appendix A), and we therefore discontinued its use for the remainder of our studies. New HP variants with longer chain length PEI (i.e., HP-10K and HP-25K) could efficiently package both 4.7 kb pmCherry-N1 and 9.3 kb pX458 Cas9 expression plasmids at N/P ratios of 1, 3, and 6; whereas those with intermediate length PEI (i.e., HP-1.8K) efficiently packaged both plasmids at high N/P ratios of 3 or 6 but not at the low N/P ratio of 1 (Figure 1E,F).

### 3.2. HP Nanoparticles Can Be Delivered into Cells

We next tested whether the developed HP nanoparticles could be delivered into various cell types. To visualize HP nanoparticles in vitro, we labeled the HP with FITC, a green fluorescent dye, prior to assembling the nanoparticles according to previous work [29]. Briefly, FITC and HP polymer in methanol were stir-mixed overnight at room temperature. The mixture was then transferred to a dialysis bag (MWCO 6–8 kDa) and dialyzed in plenty of ethanol and then water to remove any unbonded FITC. The light yellow FITC labeled HP polymer was obtained via lyophilization. We then applied the FITC labeled HP nanoparticles to Ad293 cells, followed by flow cytometry to quantify FITC signal positive cells. Surprisingly, we found that 99.5–99.9% of cells had a positive FITC signal (Figure 2A,B), implying that the HP can easily transfect Ad293 cells. We also confirmed that HP can efficiently transfect other cell types, including human (HepG2) and mouse (AML12) hepatocytes (Appendix A). To confirm that these HP nanoparticles were delivered inside the cell, not just attached to the cell surface, we dyed the cell plasma membrane with deep red and visualized the results with confocal microscopy. We found that the majority of FITC labeled HPs were localized within the cells with some co-localization with the DAPI-stained nucleus (Figure 2C). These data indicate that the HP nanoparticles can enter the cytosol of the cells and possibly the nucleus and are not simply co-localized with the membrane, a positive indication that HP can deliver pDNA into cells, a critical step towards gene editing.

### 3.3. HPs Have Low Toxicity in Various Cell Lines

Gene delivery systems utilizing PEI are known to have high cellular toxicity, but we anticipated that conjugating PEG onto the HP core would help to mitigate this ([28],[35]). We tested HP cytotoxicity in primary rabbit embryonic fibroblasts (REFs) and found that every HP formulation tested had very low cell toxicity with over 90% cell survival at an N/P ratio of 6 (Figure 3A). We also tested HP cytotoxicity in HepG2 cells and found that compared to PEI alone, HP with the same PEI chain length have much lower cytotoxicity at each polymer to nucleotide ratio tested (Figure 3B). Encouragingly, increasing the molecular weight of the PEI up to 25 kDa in the HP system did not cause an increase in cytotoxicity at an N/P ratio of 6; however, at a maximum N/P ratio of 12, the cytotoxicity did in fact increase (cell survival rate decreased to around 50% at N/P 12 for HP25 kDa). To maintain a low cytotoxicity, we used an N/P ratio of 6 in the in vitro studies, with more than 80% cell survival rates observed (Figure 3A,B).

### 3.4. pDNA Can Be Delivered by HPs and Is Expressed in Cells

To be expressed, pDNA must enter the cytosol, escape endosomal degradation, then translocate to the nucleus [34,35,36,37]. To monitor pDNA expression by transfected cells, we used red fluorescent protein (RFP) expression plasmids (5 kb) delivered by HPs to Ad293 cells. The RFP fluorescence in the cells is an indication that the pDNA was released from the HP and subsequently expressed by the cell. Flow cytometry revealed that 24% of cells transfected with RFP-pDNA using HP-25K exhibit red fluorescence after 24 h, significantly higher than those achieved by HP-1.8K and HP-10K (Figure 3C,D). Fluorescence immunostaining further confirmed that a higher percentage of cells expressed RFP in the HP-25K group than those in the HP-1.8K and HP-10K groups (Figure 3E). The higher expression is likely due to the larger size of the PEI chain in HP-25K, endowing it stronger package ability of the reporter gene pDNA (5 kb) and smaller complex size than other HPs. 

### 3.5. HP-25K Transfection Efficiency Is Higher than Commercial Transfection Reagents Lipofectamine and PEI25K in HepG2 and REF Cells

After confirming that HP-25K was the most efficient of the HPs tested in transfection of Ad293 cells, we next sought to answer whether it was better than the commercially available transfection reagents Lipofectamine and PEI25K. We transfected green fluorescent protein (GFP) expression plasmids (6.1 kb) into HepG2 cells using Lipofectamine, PEI25k, and HP-25k. We found that HP-25K has a transfection efficiency about two times higher than both PEI25K and Lipofectamine (Figure 3F,H). In REF cells, HP-25K was also the most effective transfection reagent, leading to more than twice the GFP expression of Lipofectamine (Figure 3G,I). We repeated this experiment in HeLa cells, Ad293 cells, and AML12 cells, and found that in all cell types, HP-25K had a higher or comparable transfection efficiency to Lipofectamine or PEI25K (Appendix A). Taken together, these data prove that HP-25K nanoparticles generally perform better than the commercially available Lipofectamine or PEI25K, in some cases doubling the efficiency of Lipofectamine and PEI25K.

### 3.6. HP-25K Mediated Delivery of Cas9 pDNA Induces Indel Mutations in Target Sites In Vitro

To test whether HP-25K can efficiently deliver the Cas9 pDNA and introduce indels in target cells, we designed two guide RNAs (gRNAs) TERT-sg1 and TERT-sg2 (Figure 4A) both targeting the telomere reverse transcriptase (TERT), the gene coding for the enzymatic subunit of telomerase [38]. Ad293 cells were transfected with pX330 pDNA which expresses Cas9 and customized gRNA (e.g., TERT-sg1 or TERT-sg2) by HP-25K or Lipofectamine. Genomic DNAs were extracted two days post transfection for indel analysis.

The T7E1 assay indicated that the HP-25K mediated delivery of pX330 pDNA led to similar or higher indel mutations at the target sites than Lipofectamine (Figure 4B). The findings were confirmed by Deepseq, with HP-25K achieving similar or higher indel rates (25.35% and 41.82%) than Lipofectamine (24.47% and 27.17%) for TERT-sg1 and TERT-sg2, respectively (Figure 4C).

We then tested targeting a new locus, MYLIP, in Ad293 and HepG2 cells, by using HP-25K or Lipofectamine to deliver PX458, another commonly used Cas9 expressing pDNA of larger size (9.3 kb) than PX330 (8.5 kb) [32]. Consistently, HP-25K led to higher indel rates (9.09% and 4.88%) than Lipofectamine (4.01% and 0.91%) for Ad293 and HepG2, respectively (Figure 4C).

These data indicate that our newly developed HP-25K successfully and efficiently delivered Cas9 plasmids up to 9.3 kb into different cell types to achieve efficient genome editing at different loci.

### 3.7. HP-25K Mediated Delivery of BE4-Max pDNA Leads to Base Editing in Ad293 Cells

Given the success of the HP system for delivery of Cas9 pDNAs, we reasoned that this system can be readily used to deliver a Cas9 variant BE4-max pDNA (pCMV-BE4max, Addgene #112093, 8.96 kb) [39]. Base editors (BEs) are gaining momentum in both basic and translational research because they achieve base editing without creating double stranded breaks (DSB) which many consider as a safer alternative to the DSB based Cas9 genome editing; however, BEs also require an effective delivery system to accelerate their impact on clinical therapy [40]. To test this, we designed gRNA (Figure 4D) targeting the PDCD1 gene, which encodes for the key protein PD-1 which is of clinical significance in cancer immunotherapy [41], and used HP-25K to deliver pCMV-BE4max (BE4) to Ad293 cells. Robust on-target base editing rates (9.99%) were achieved by HP-25K, approximately doubling those by Lipofectamine (5.18%) (Figure 4D). While future work is necessary to further verify advantages of HP-25K over Lipofectamine, the results demonstrate that HP-25K can deliver pDNA encoding BE nucleases of up to 8.96 kb various cells, enabling efficient base editing. To our knowledge, this is the first report of nanoparticle mediated delivery of based editor elements to human cells.

Here, we show that HP-25K can deliver Cas9 and BE4 pDNAs and induce mutations with low cytotoxicity, demonstrating the potential of this system in gene editing applications. There is still need and opportunity to improve the efficiency of the HP system which will require further optimization of the HP itself and the release of pDNA from the vector. The robust pDNA expression achieved in this work could be further improved by altering the HP in several ways. The unique advantage of HP-25K is the engineering potential of the HP core, PEG, and PEI chains, which have ample space and functionalities to bind various components. By optimizing the ratio between PEI and PEG attached to the HP core, the next generation of HP nanoparticles may be able to achieve a higher transfection efficiency and lower cytotoxicity than HP-25K. Another engineering opportunity is to integrate targeting elements, such as beta-d-galactose for hepatocyte targeting, into the HP system to enable tissue specific delivery and improve on-target efficiency and reduce off-target side effects. To attain the high in vivo efficacy required of gene-editing based systems, we imagine that the HP-pDNA polyplexes may require a sustained release not achieved with polyplexes alone. Previous work from our lab has shown that increased duration of poly(lactic-*co*-glycolic acid) (PLGA) microsphere release can enhance the in vivo stability of miRNA delivery systems [8,25].

An interesting observation from the present study is that although nearly all cells exhibit uptake of the polyplexes, robust reporter gene expression is only found when HP-25K is used. This may indicate that there is a disparity between the number of polyplexes getting into the cell and the amount of pDNA being expressed. A possible explanation is that not all nanoparticles were able to undergo endosomal escape once inside the cell, decreasing the observed expression. We reason that such endosomal escape efficiency is dependent upon the charge of the PEI, and therefore the PEI MW. HP-25K is optimized for condensation with pDNA, so the PEI chain length should not be changed, but it may be helpful to add an endosome disruptor to the HP core to facilitate intracellular release of the cargo. In this way, gene editing efficiencies achieved via HP-25K mediated delivery could be further improved.

## 4. Conclusions

Gene editing nucleases such as Cas9 have become important tools in biomedical research, offering new possibilities to cure or treat monogenetic diseases (e.g., Duchenne muscular dystrophy, Cystic Fibrosis, etc.), and other complex diseases (e.g., AIDS, dyslipidemia, cancer, etc.) [25]. The clinical application of gene editing therapy requires both efficient and safe delivery of nuclease elements. While more efficient, viral mediated Cas9 pDNA delivery has demonstrated immunogenicity, insertional mutagenesis, and size limitations for DNA cargo [42].

The present work demonstrates that HP-25K can safely and efficiently transfect multiple cell types with pDNAs of various sizes with low cytotoxicity. Importantly, this system enables Cas9 pDNA delivery to several clinically relevant cell lines, and results in delivery efficiency rivaling and surpassing the commercially available Lipofectamine and PEI25K even for pDNA up to 9.3 kb. HP delivery of pDNA encoding gene editing nucleases can induce indel mutations and efficient BE gene editing in human cells. While further work remains to be done in optimizing the HP nanoparticle system, HPs represent a promising vehicle for the non-viral delivery of CRISPR/Cas9 pDNAs towards the clinical application of gene-editing based therapy.

## Figures and Tables

**Figure 1 cells-12-00156-f001:**
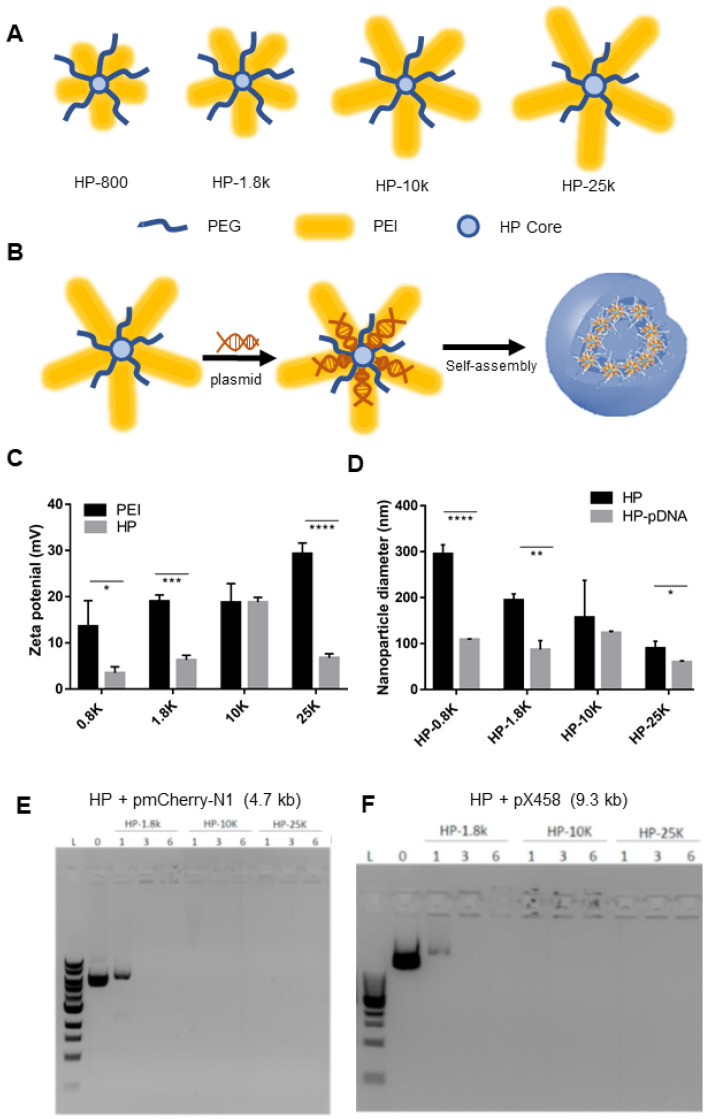
HP nanoparticle design and characterization. (**A**) Illustration of the design of HP- nanoparticles. Blue arm is PEG, yellow is PEI (larger PEI shape referring to higher PEI molecular weight), light blue is HP core. (**B**) Illustration of the double shelled polyplexes formed by HP nanoparticles and pDNA. (**C**) Zeta potential of HP nanoparticles and PEI of the same chain lengths. (**D**) Diameters of HP nanoparticles with or without packaging pmCherry-N1 (4.7 kb) pDNAs determined by Dynamic Light Scattering. (**E**) Gel retardation assay of HP and pmCherry-N1 (4.7 kb) polyplexes at various N/P ratios (1, 3, 6). (**F**) Gel retardation assay of HP and pX458 (9.3 kb) polyplexes at various N/P ratios (1, 3, 6). * *p* < 0.05. ** *p* < 0.01. *** *p* < 0.001. **** *p* < 0.0001.

**Figure 2 cells-12-00156-f002:**
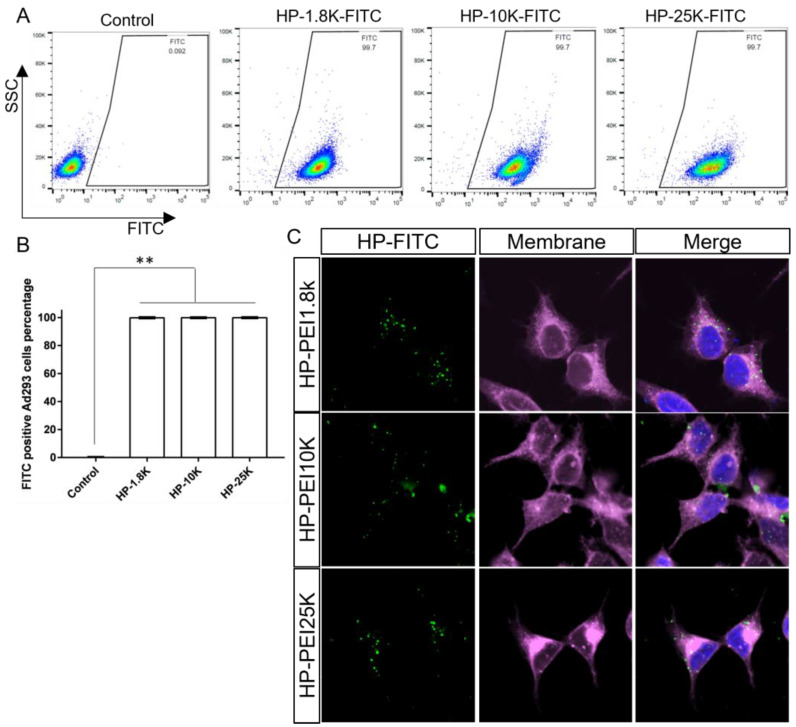
HP nanoparticles enter cells efficiently. (**A**) Representative flow cytometry results of Ad293 cells treated with FITC labeled HP nanoparticles. (**B**) Quantitative summary of flow cytometry results. (**C**) Confocal microscopy images of Ad293 cells treated with FITC labeled HP nanoparticles. Green: FITC labeled HP nanoparticles. Purple: membranes. Blue: nuclei. ** *p* < 0.01.

**Figure 3 cells-12-00156-f003:**
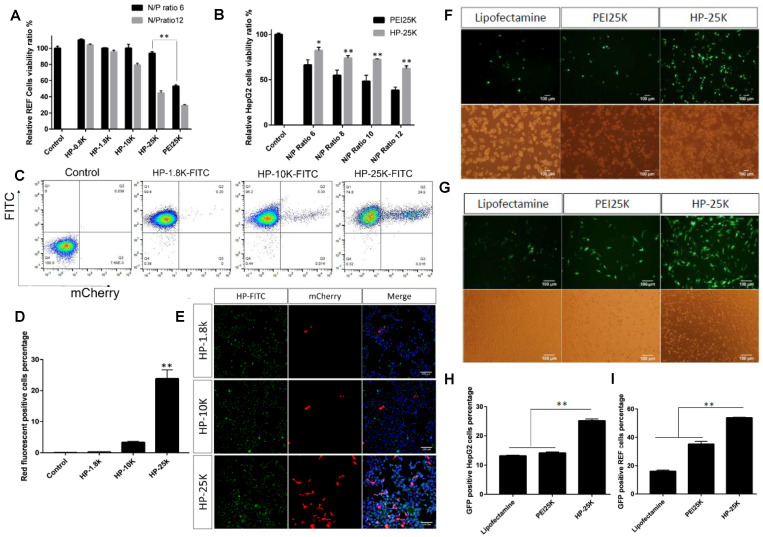
HPs safely and efficiently deliver fluorescence reporter gene pDNAs to various cell types. (**A**) REF cell survivability after transfection with different size HP-pDNA polyplexes. (**B**) HepG2 cell survivability after transfection with HP-25K or PEI25K -pDNA polyplexes at different N/P ratios. (**C**) Representative flow cytometry results of Ad293 cells transfected with HP-pmCherry-N1 polyplexes. (**D**) Quantitative summary of flow cytometry results. (**E**) Confocal microscopy images of Ad293 cells transfected with HP-pmCherry-N1 polyplexes. Green: FITC-labeled HP nanoparticles. Red: mCherry fluorescence. Blue: nuclei. Scale bar 100 μm. (**F**) Microscopy images of HepG2 cells transfected with HP-eGFP polyplexes. Top: fluorescence microscopy images. Green: eGFP fluorescence. Bottom: light microscopy images. (**G**) Microscopy images of REF cells transfected with HP-eGFP polyplexes. Top: fluorescence microscopy images. Green: eGFP fluorescence. Bottom: light microscopy images. (**H**) Quantification of flow cytometry results of HepG2 cells transfected with HP-eGFP polyplexes. (**I**) Quantification of flow cytometry results of REF cells transfected with HP-eGFP polyplexes. * *p* < 0.05. ** *p* < 0.01.

**Figure 4 cells-12-00156-f004:**
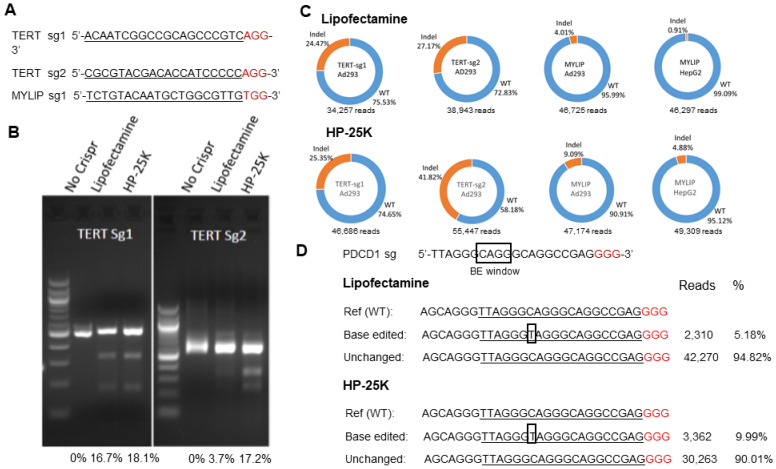
HP-25K nanoparticle mediated delivery of Cas9 and BE pDNAs to cells leads to efficient gene editing. (**A**) List of gRNAs used with Cas9 pDNAs. (**B**) T7E1 assay on Ad293 cells transfected with Cas9 pDNAs targeting TERT gene by HP-25K or Lipofectamine. (**C**) Deepseq analysis of gene editing events in Ad293 and HepG2 cells targeting TERT or MYLIP by HP-25K or Lipofectamine. (**D**) Deepseq analysis of BE events in Ad293 cells transfected with BE4-max pDNAs targeting PDCD1 by HP-25K or Lipofectamine. Boxed letters in the PDCD1 sg sequence indicate the predicated BE window. Boxed letters in the Deepseq results indicate the detected base changes. Reads are total number of counts by Deepseq.

## Data Availability

All data are available in the Figures and Appendix A.

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
