# Peer review of "Delivery of CRISPR/Cas9 Plasmid DNA by Hyperbranched Polymeric Nanoparticles Enables Efficient Gene Editing"

_cells, 2022, doi:10.3390/cells12010156_

Round 1
Reviewer 1 Report
This is very interesting in our readers due to CRISPR/Cas9 engineering of cells and prganisms.
Although the CRISPR/Cas9 technology has well been evolved in the current research application, the present hyperbranched polymeric nanoparticles-based delivery of edited gene is to documented in journals. Therefore, Cells journal has a medium to document the present study.
In a minor comment, no comprehensively comparative literatures of the vairous nanoparticles are described.
Author Response
Comments:
Although the CRISPR/Cas9 technology has well been evolved in the current research application, the present hyperbranched polymeric nanoparticles-based delivery of edited gene is to documented in journals. Therefore, Cells journal has a medium to document the present study.
In a minor comment, no comprehensively comparative literatures of the vairous nanoparticles are described.
Response: Thank you for the comments. We agree with the reviewer and added more information of non-viral gene vectors and nanoparticles for gene delivery in the introduction section (changes are tracked in the revised manuscript): “Although liposome and polymer based gene delivery has made significant progress in in-vitro studies and has received deserved attention during the past two decades, these non-viral gene delivery systems are still facing challenges when applied in-vivo such as rapid clearance from blood, toxicity, and low efficiency [22,23]. A vast improvement in safety and efficiency for in vivo gene delivery systems is required for clinical application.”
Reviewer 2 Report
The study by Xiu and colleagues is compact, well-designed and executed to demonstrate the delivery potential of the developed by this group hyperbranched polymer (HP) system of delivery for plasmid DNA, best performing in the study as 25K particles. My comments are mainly editorial; however, I have some requests for more specifications in the text and also some questions which may require commenting also in the manuscript.
Questions:
1. What is the effect of DNA quality (e.g. A260/280 ratio, % of ccc form), was pDNA always checked before polyplex formation, and was is the same pDNA preparation used throughout the study?
2. Figure 1. The idea of the scheme in A. was to demonstrate the length of the PEI chains; however, using this way of depiction, it points at the increasing diameter of the HP nanoparticle with the PEI length. The DLS analysis of diameter points at the opposite, and then HP nanoparticles with DNA seem to have similar size, with some oscillations, like 10k being the largest (120 nm) and 25k the smallest (75 nm?). Could you comment on the size of the polyplexes in the paper and explain? Unfortunately, only HP-25k polyplex is depicted in TEM in S1. The size of the HP polyplexes should be important as regards their internalization.
3. Line 283: allowing it to package more of the reporter gene pDNA (5kb) than other HPs: this does not seem to correspond with the gel retardation assay (even though encapsulation may inhibit dye incorporation). Please give experimental grounds for such a statement. Can better uptake correlate with a bit smaller size of the particles for 25K? The discussion points to the differences in endosomal release.
4. How repeatable is HP Solution preparation, e.g., measured by DLS? This is important when thinking about its application.
5. Could you comment on whether HP delivery can be “addressable” to obtain delivery to desired tissues (e.g. as regards cancer cells).
6. FITC labeling of HP nanoparticles should be briefly described in Methods without the necessity to go to reference 15. What kind of analysis was used to analyze whether polyplexes with different PEI lengths incorporate FITC to the same extent (the fluorescent signal of the particles used in the uptake assay is the same)?
Editorial/specification requests:
1. Line 63 and 126: Unfortunately, time flies and 2016 is already not a “recent” time (even more 2009 quoted as recent in line 126), I would use Previously instead Recently.
2. Instead of devoting a whole paragraph on explanation why viral vectors are not the best option for delivery of CRISPR/Cas9 components, the authors could briefly summarize other non-viral delivery approaches.
3. Methods: I recommend moving 2.6 Plasmid and Cas9 construct preparation before section 2.3 Particle Size and Zeta Potential Measurements as the plasmids appear in 2.3 and this way they are mentioned for the first time without proper description.
4. Although the paper is generally carefully written as regards corrections, some editorial mistakes are still present: spaces between amounts and units, like 6ml- 6 ml, 20nm, etc., some letters missing (line 129: described), line 144 70um instead of micro, line 214-215 use “were” instead of “are”.
5. Line 241: I cannot find the description of Ad293 cells in the Methods.
6. Figure 2C: Just a comment: if you use confocal imaging, the best way to demonstrate the intracellular distribution of the particles would be Z-stacking and their reconstitution to get 3D-images.
7. Line 268: “ .. the cytotoxicity did in fact increase”. Please state up to which value it increased (otherwise it seems manipulative). Please state in the Methods what was the time of cytotoxicity analysis after uptake (24h?).
8. Line 276: “Flow cytometry revealed that 24% of cells transfected with RFP‐pDNA using HP‐25K exhibit red fluorescence”. Please state what was the time of expression, 24h?
Author Response
Comments:
The study by Xiu and colleagues is compact, well-designed and executed to demonstrate the delivery potential of the developed by this group hyperbranched polymer (HP) system of delivery for plasmid DNA, best performing in the study as 25K particles. My comments are mainly editorial; however, I have some requests for more specifications in the text and also some questions which may require commenting also in the manuscript.
Questions:
- What is the effect of DNA quality (e.g. A260/280 ratio, % of ccc form), was pDNA always checked before polyplex formation, and was is the same pDNA preparation used throughout the study?
Response: The pDNA was extracted with a “QIAGEN Plasmid Maxi Kit” and the A260/280 ratio was about 1.8. We made sure that all pDNAs used in this study were prepared with the same procedure and quality was checked before used. The plasmid quality information was also added in the content.
- Figure 1. The idea of the scheme in A. was to demonstrate the length of the PEI chains; however, using this way of depiction, it points at the increasing diameter of the HP nanoparticle with the PEI length. The DLS analysis of diameter points at the opposite, and then HP nanoparticles with DNA seem to have similar size, with some oscillations, like 10k being the largest (120 nm) and 25k the smallest (75 nm?). Could you comment on the size of the polyplexes in the paper and explain? Unfortunately, only HP-25k polyplex is depicted in TEM in S1. The size of the HP polyplexes should be important as regards their internalization.
Response: We are sorry for the confusion in the scheme. The yellow colored PEI shape in figure 1A refers to the molecular weight change of the PEI rather than the particle size change. We added the explanation in the caption: “yellow is PEI (larger PEI size referring to higher PEI molecular weight)” to address the reviewer’s concern.
As for the cationic polymer, before complexing with genes, the molecular structure is stretched in the aqueous solution due to the repulsion between the abundant positive changes. However, when complexed with genes, the net positive charges are decreased since the negatively charged genes neutralized the positive charges. As a result, the complex shows a shrunk state in aqueous solution and smaller size than that of the cationic polymer. So, we added the following sentence to summarize the size change: “The observation that the size of the complex is smaller than that of the HP polymer is in good agreement with the literature [33].” We also added “In summary, all HP/pDNA complexes showed the particle size to be less than 150 nm.” to enphasize that the size range of the HP/DNA complexes is good for transfection purpose (in the nanometer size (ref: 33)).
- Line 283: allowing it to package more of the reporter gene pDNA (5kb) than other HPs: this does not seem to correspond with the gel retardation assay (even though encapsulation may inhibit dye incorporation). Please give experimental grounds for such a statement. Can better uptake correlate with a bit smaller size of the particles for 25K? The discussion points to the differences in endosomal release.
Response: We agree with the reviewer that this statement is very general and is not very precise. We modified this sentence according to the reviewer’s comment, so the statement is more appropriate. Here is the revised text: “The higher expression is likely due to the larger size of the PEI chain in HP-25K, endowing it stronger package ability of the reporter gene pDNA (5kb) and smaller complex size than other HPs”.
- How repeatable is HP Solution preparation, e.g., measured by DLS? This is important when thinking about its application.
Response: The HP polymers were stored as dry powders and the solutions were freshly prepared each time for the experiments (such as DLS, transfection, or agarose gel electrophoresis). And in this way, all the experiments are repeatable.
- Could you comment on whether HP delivery can be “addressable” to obtain delivery to desired tissues (e.g. as regards cancer cells).
Response: This is a good comment. At present, the HP polymers have no specificity to transfect certain type of cells, tissues, or organs. However, our team is working on the targeted gene delivery via chemical modification of the HP polymers.
- FITC labeling of HP nanoparticles should be briefly described in Methods without the necessity to go to reference 15. What kind of analysis was used to analyze whether polyplexes with different PEI lengths incorporate FITC to the same extent (the fluorescent signal of the particles used in the uptake assay is the same)?
Response: Thank you for pointing this out. We added the following sentences to introduce the FITC labeling process: “Briefly, FITC and HP polymer in methanol were stir-mixed overnight at room temperature. The mixture was then transferred to a dialysis bag (MWCO 6-8 kDa) and dialyzed in plenty of ethanol and then water to remove any unbonded FITC. The light yellow FITC labeled HP polymer was obtained via lyophilization.” We used NMR to confirm the FITC peak in the FITC labeled HP. We fully agree and understand the reviewer’s concern of the FITC extent. We set the FITC:HP polymer ratio of 1:1 (moles: moles) when we did the labeling to make sure all labeled polymer are under the same standard.
Editorial/specification requests:
- Line 63 and 126: Unfortunately, time flies and 2016 is already not a “recent” time (even more 2009 quoted as recent in line 126), I would use Previously instead Recently.
Response: Thank you for pointing out. We have done the change accordingly.
- Instead of devoting a whole paragraph on explanation why viral vectors are not the best option for delivery of CRISPR/Cas9 components, the authors could briefly summarize other non-viral delivery approaches.
Response: More introductory information was added to highlight the reason why we are focusing on the improvement of polymer nanoparticles for gene delivery applications: “Although liposome and polymer based gene delivery has made significant progress in in-vitro studies and has received deserved attention during the past two decades, these non-viral gene delivery systems are still facing challenges when applied in-vivo such as rapid clearance from blood, toxicity, and low efficiency. A vast improvement in safety and efficiency for in vivo gene delivery tools systems is required to guaranteefor clinical application”.
- Methods: I recommend moving 2.6 Plasmid and Cas9 construct preparation before section 2.3 Particle Size and Zeta Potential Measurements as the plasmids appear in 2.3 and this way they are mentioned for the first time without proper description.
Response: Thank you for the suggestion. We made the changes accordingly.
- Although the paper is generally carefully written as regards corrections, some editorial mistakes are still present: spaces between amounts and units, like 6ml- 6 ml, 20nm, etc., some letters missing (line 129: described), line 144 70um instead of micro, line 214-215 use “were” instead of “are”.
Response: Thank you. We corrected these errors as suggested.
- Line 241: I cannot find the description of Ad293 cells in the Methods.
Response: The Ad293 cell information was added in the method section: “Cells (Ad293 (a derivative of HEK293), human (HepG2) or mouse (AML12)) were seeded in a 24-well plate one day before the assay”.
- Figure 2C: Just a comment: if you use confocal imaging, the best way to demonstrate the intracellular distribution of the particles would be Z-stacking and their reconstitution to get 3D-images.
Response: Thank you for the suggestion. We will try this method in the future.
- Line 268: “ .. the cytotoxicity did in fact increase”. Please state up to which value it increased (otherwise it seems manipulative). Please state in the Methods what was the time of cytotoxicity analysis after uptake (24h?).
Response: Thank you. We added the cytotoxicity information in the content: “(cell survival rate decreased to around 50% at N/P 12 for HP25 kDa)”.
- Line 276: “Flow cytometry revealed that 24% of cells transfected with RFP‐pDNA using HP‐25K exhibit red fluorescence”. Please state what was the time of expression, 24h?
Response: Thank you. We added the time of expression: “Flow cytometry revealed that 24% of cells transfected with RFP-pDNA using HP-25K exhibit red fluorescence after 24 hours”.
Reviewer 3 Report
The authors reported the design and synthesis of hyperbranched polymers made of different molecular weight PEI chains and analyzed their gene packaging activity, cytotoxicity and transfection efficacy. To improve the gene packaging capacity, the MW of PEI was increased while the same composition of PEG and HP was kept. The work is interesting and addresses the issue of using nanoparticles as non-viral vehicles for transfecting large plasmids, in particular CRISPR/Cas9 pDNAs . However, there are some problems listed as follows:
1) From flow cytometry experiment to quantify FITC signal positive cells shown in Figure 2A and B, the authors claim that 99,9% of cells have NP nanoparticles inside. This conclusion is not convincing. Ctrl cells do not show fluorescence, but it is not indicated whether these cells have been incubated with FITC! Please define the negative control. How can the authors exclude that the fluorescence signal is due to free FITC molecules extracellularly or intracellularly released during the time of the experiment? If only vectors are labeled with FITC, what is the proof that intact HP nanoparticles, including plasmids, have entered inside the cells?
2) The three panels HP-FITC of Figure 2C are of low quality. They are almost completely black and it is impossible to detect the intracellular green fluorescence.
3) The protocol for labeling NPs with FITC is not described. On page 6 is written: “…we labeled the HP with FITC … according to previous work (15)”, but reference 15 (Nguyen et al. 2009) is a review and there are no experimental details on HP labelling with FITC.
4) The origin, characteristics and cell culture media for all different cell types used in the paper, such as Ad293 cells, primary rabbit fibroblasts, HepG2, AML12 and HeLa, must be briefly described in Material and Methods.
5) Transfection protocols for Lipofectamine, PEI25K and HP25K must be briefly described.
6) HDR abbreviation should be explained.
7) Details on statistic methods used and number of replications done for all experiments are missing and must be added.
Author Response
Comments:
The authors reported the design and synthesis of hyperbranched polymers made of different molecular weight PEI chains and analyzed their gene packaging activity, cytotoxicity and transfection efficacy. To improve the gene packaging capacity, the MW of PEI was increased while the same composition of PEG and HP was kept. The work is interesting and addresses the issue of using nanoparticles as non-viral vehicles for transfecting large plasmids, in particular CRISPR/Cas9 pDNAs . However, there are some problems listed as follows:
1) From flow cytometry experiment to quantify FITC signal positive cells shown in Figure 2A and B, the authors claim that 99,9% of cells have NP nanoparticles inside. This conclusion is not convincing. Ctrl cells do not show fluorescence, but it is not indicated whether these cells have been incubated with FITC! Please define the negative control. How can the authors exclude that the fluorescence signal is due to free FITC molecules extracellularly or intracellularly released during the time of the experiment? If only vectors are labeled with FITC, what is the proof that intact HP nanoparticles, including plasmids, have entered inside the cells?
Response: Thank you for the comments. We would like to start with the process of FITC labled HP poylmer to answer this question. The HP polymer was labeled with FITC and the products were dialyzed against ethanol and then water to remove any residule free FITC. The dialysis was carried out with plenty of solvent (more than 100 times volume than the HP polymer solution) and eventually the solvent became clear and colorless, confirming complete removal of free FITC. The HP polymer was obtained via freeze drying and was light yellow. In this way, we ensured that HP polymers were labeled without residual free FITC [ref: 29]. We also used agarose gel electrophoresis to confirm that the FITC labeled HP could successfully package plasmid DNA. Hereby, we could track the HP/pDNA complex by tracking FITC. For the flow cytometry experiment, we applied the FITC labeled HP nanoparticles to the cells, and we used normal cells (without HP nanoparticle treatment) as the control. We harvested the cells, washed cells with PBS, and resuspended the cells in PBS as required by the technician from the flow cytometry core. During this process, any FITC-HPs that are not attached to the cells are removed. As a result, the FITC positive cells should be transfected by the FITC-HP nanoparticles. We agree with the reviewer about the statement of “99,9% of cells have NP nanoparticles inside.” The word “inside” is not precise because the flow cytometry experiment can only tell if there is FITC but can not tell if the FITC is “inside” the cell. And this was the reason why we used figure 2C to identify that the FITC was inside the cell. We have modified this sentence into: “Surprisingly, we found that 99.5-99.9% of cells had a positive FITC signal (Figure 2A & 2B), implying that the HP can easily transfect Ad293 cells. We also confirmed that HP can efficiently transfect other cell types, including human (HepG2) and mouse (AML12) hepatocytes (Supplementary Figure 3).”
2) The three panels HP-FITC of Figure 2C are of low quality. They are almost completely black and it is impossible to detect the intracellular green fluorescence.
Response: An updated high-resolution Figure 2C has replaced the old one in the revision.
3) The protocol for labeling NPs with FITC is not described. On page 6 is written: “…we labeled the HP with FITC … according to previous work (15)”, but reference 15 (Nguyen et al. 2009) is a review and there are no experimental details on HP labelling with FITC.
Response: Thank you for pointing this out. The correct reference (ref 29) was cited, and the following labeling protocol was added in the content: “Briefly, FITC and HP polymer in methanol were stir-mixed overnight at room temperature. The mixture was then transferred to a dialysis bag (MWCO 6-8 kDa) and dialyzed in plenty of ethanol and then water to remove any unbonded FITC. The light yellow FITC labeled HP polymer was obtained via lyophilization.”
4) The origin, characteristics and cell culture media for all different cell types used in the paper, such as Ad293 cells, primary rabbit fibroblasts, HepG2, AML12 and HeLa, must be briefly described in Material and Methods.
Response: Thank you. The following information was added to address this concern: “Cells: Ad293 cells (Catalog# 240085) were obtained from Agilent. HepG2 cells (Catalog# HB-8065) and Hela cells (Catalog# CCL-2) were obtained from ATCC. Ad293, HepG2 and Hela cells were cultured in Dulbecco's Modified Eagle Medium (DMEM, Catalog#11965092, ThermoFisher) supplemented with 10% fetal bovine serum (FBS, Catalog# SH30071.03HI, Hyclone) and 1% penicillin-streptomycin (Catalog# 15140122, ThermoFisher). AML-12 cells (Catalog# CRL-2254) were purchased from ATCC, and cultured in DMEM:F12 (Catalog# 11320033, ThermoFisher) supplemented with 10%FBS, 1% insulin-transferrin-selenium (ITS, Catalog# 41400045, ThermoFisher), 40ng/mL dexamethasone (Catalog# D4902, Sigma-Aldrich), and 1% penicillin-streptomycin(Catalog# 15140122, ThermoFisher). Rabbit fibroblast cells were established in house from ear biopsies collected from New Zealand White Rabbits. The ear biopsy procedure is approved by the University of Michigan Institutional Animal Care and Use Committee (IACUC) protocol #PRO00010094. Rabbit fibroblast cells were cultured in DMEM (Catalog#11965092, ThermoFisher) supplemented with 10% FBS (Catalog# SH30071.03HI, Hyclone), and 1% penicillin-streptomycin (Catalog# 15140122, ThermoFisher).”
5) Transfection protocols for Lipofectamine, PEI25K and HP25K must be briefly described.
Response: The following change was made to provide more detailed transfection process: “Cells (Ad293 (a derivative of HEK293), human (HepG2) or mouse (AML12)) were seeded in a 24-well plate one day before the assay. Upon transfection, 1 µg of pDNA and polymer (HPs or PEIs) at the desired N/P ratio (w/w) were diluted in 50 µl of Opti-MEM (#31985062, Invitrogen), incubated at room temperature for 30 minutes, and subsequently added into each well of cell culture. The control (lipofectamine) transfection process was following the manufacture’s protocol. Briefly, 50 µl lipofec-tamine dilution in Opti-MEM and 50 µl DNA dilution in Opti-MEM were mixed and incubated for 15 minutes. Then, the mixture was added into the well of cell culture. The transfection result was obtained after 24 hours.”
6) HDR abbreviation should be explained.
Response: Thank you for poiting this out. We added the full name of HDR (Homology directed repair) in the content.
7) Details on statistic methods used and number of replications done for all experiments are missing and must be added.
Response: The statistics information was added:” 2.14 Statistics
Data are expressed as mean + standard error of means (SEM) from three replicates in bar graphs, and were analyzed and compared using unpaired, 2-tailed Student’s t test (Graphpad Prism 9.2.0, San Diego, CA, USA). Statistical significance with P<0.05 is considered significant. Different levels of significant statistical differences were indicated by number of * in each graph, where *P<0.05, **P<0.01, and ***P<0.001.”
Round 2
Reviewer 3 Report
The manuscript can be accepted in the present form.